# Geometric characterization of anomalous Landau levels of isolated flat bands

Yoonseok Hwang [1,2,3], Jun-Won Rhim [1,2,4✉] & Bohm-Jung Yang [1,2,3✉]

According to the Onsager's semiclassical quantization rule, the Landau levels of a band are bounded by its upper and lower band edges at zero magnetic field. However, there are two notable systems where the Landau level spectra violate this expectation, including topological bands and flat bands with singular band crossings, whose wave functions possess some singularities. Here, we introduce a distinct class of flat band systems where anomalous Landau level spreading (LLS) appears outside the zero-field energy bounds, although the relevant wave function is nonsingular. The anomalous LLS of isolated flat bands are governed by the cross-gap Berry connection that measures the wave-function geometry of multi bands. We also find that symmetry puts strong constraints on the LLS of flat bands. Our work demonstrates that an isolated flat band is an ideal system for studying the fundamental role of wave-function geometry in describing magnetic responses of solids.

[1] Center for Correlated Electron Systems, Institute for Basic Science (IBS), Seoul 08826, Korea. [2] Department of Physics and Astronomy, Seoul National University, Seoul 08826, Korea. [3] Center for Theoretical Physics (CTP), Seoul National University, Seoul 08826, Korea. [4] Department of Physics, Ajou University, Suwon 16499, Korea. ✉email: jwrhim@ajou.ac.kr; bjyang@snu.ac.kr

The geometry of Bloch wave functions, manifested in the quantities such as Berry curvature and Berry phase, is a central notion in the modern description of condensed matter. Due to the significant role of wave-function geometry in describing the fundamental properties of solids, finding efficient methods of measuring it has been considered a quintessential problem in solid-state physics. In this respect, examining the Landau level spectrum has received particular attention, as one of the most efficient and convenient methods for detecting the geometric properties of Bloch states.

A conventional way of determining the Landau levels of Bloch states is to use the semiclassical approach based on Onsager's semiclassical quantization rule given by

$$S_0(\epsilon) = \frac{2\pi eB}{\hbar}\left(n + \frac{1}{2} - \frac{\gamma_{\epsilon,B}}{2\pi}\right), \quad (1)$$

which is generally valid in the weak-field limits. Here $S_0(\epsilon)$ is the area of a closed semiclassical orbit at the energy $\epsilon$ in momentum space, $B$ is a magnetic field, $e$ is the electric charge, $\hbar$ is the reduced Planck constant, and $n$ is a non-negative integer. The last term $\gamma_{\epsilon,B}$ indicates the quantum correction from Berry phase, orbital magnetization, etc.[1–5], reflecting the geometric properties of solids. A collection of discrete energies ($\epsilon$) satisfying Eq. (1) forms the Landau levels which critically depend on the geometric quantity $\gamma_{\epsilon,B}$. For instance, in graphene with relativistic energy dispersion, Eq. (1) successfully predicts the $\sqrt{nB}$ dependence of the Landau levels, where the existence of the zero-energy Landau level is a direct manifestation of the $\pi$-Berry phase of massless Dirac particles[6, 7]. Later, this semiclassical approach is generalized further to the cases with an arbitrary strength of magnetic field[8] where the zero-field energy dispersion in Eq. (1) is replaced by the magnetic band structure with $B$ linear quantum corrections.

The Onsager's semiclassical scheme has provided a powerful method of understanding complicated Landau level spectra of solids intuitively. In usual dispersive bands where $B$ linear quantum corrections are negligible in weak-field limit, Onsager's semiclassical approach in Eq. (1) predicts that the Landau levels are developed in the energy interval bounded by the upper and lower band edges of the zero-field band structure. However, there are a few examples of violating this expectation. Especially, several systems exhibit anomalous Landau levels appearing in gapped regions away from the zero-field energy bounds where the semiclassical orbit, as well as $S_0(\epsilon)$, cannot be defined, according to Eq. (1). One famous example is the Landau levels of a Chern band which appear in an adjacent energy gap at zero-field. Similar behavior was also recently predicted in fragile topological bands characterized by nonzero Euler numbers[9–11]. More recently, it was shown that anomalous Landau levels also appear in singular flat bands[12, 13], where a flat band is crossing with another parabolic band at a momentum[14]. Interestingly, it is found that the Landau levels of a singular flat band appear in the energy region with a vanishing density of states at zero magnetic fields. Moreover, the total energy spreading of the flat band's Landau levels, dubbed the Landau level spreading (LLS), is solely determined by a geometric quantity, called the maximum quantum distance which characterizes the singularity of the relevant Bloch wave function[14].

In this work, we propose a distinct class of flat-band systems that exhibit anomalous Landau level structures. The flat band we consider is isolated from other bands by a gap, which we call an isolated flat band (IFB). An IFB is generally non-singular as well as topologically trivial[15–17] as opposed to nearly flat topological bands or degenerate flat bands[18, 19], so that it does not belong to any category of the systems exhibiting anomalous Landau levels discussed above. However, it is found that the Landau levels of IFBs are anomalous, that is, unbounded by the original band structure at zero magnetic fields and developed in the band gaps above and below the flat band.

In fact, the Onsager's semiclassical quantization rule in Eq. (1) generally does not work in flat bands, unless the $B$ linear quantum corrections are properly included. This is because there are infinitely many semiclassical orbits allowed so that $S_0(\epsilon)$ cannot be uniquely determined. Interestingly, after taking into account the $B$ linear quantum corrections, we find that an IFB generally exhibits anomalous LLS, and the upper and lower energy bounds for the LLS are determined by the cross-gap Berry connection defined as

$$A_i^{nm}(\mathbf{k}) = i\langle u_n(\mathbf{k})|\partial_i u_m(\mathbf{k})\rangle \quad (n \neq m), \quad (2)$$

where $u_n(\mathbf{k})$ is the periodic part of the Bloch wave function of the $n$th band[20]. This is a multi-band extension of the conventional Abelian Berry connection and describes inter-band couplings. Let us note that, unlike the Abelian Berry connection defined for a single band, the cross-gap Berry connection $A_i^{nm}(\mathbf{k})\,(n \neq m)$ is gauge-covariant. We will show that the LLS of an IFB is given by the product of the $x$ and $y$ components of the cross-gap Berry connection between the flat band and other bands weighted by their energy. The LLS of an IFB is strongly constrained by the symmetry of the system, which is demonstrated in various flat band models including the Lieb and the Tasaki models as well as the model describing twisted bilayer graphene (see the "Results" section and Supplementary Note 4). Our work demonstrates the fundamental role of wave-function geometry in describing the Landau levels of flat bands.

## Results

**Modified band dispersion and the LLS.** The original Onsager's semiclassical approach predicts IFBs inert under external magnetic field, and thus it cannot explain the LLS of IFBs. On the other hand, the *modified semiclassical approach* developed by M.-C. Chang and Q. Niu[8] can resolve this problem. Contrary to the Onsager's approach, where the band structure at zero magnetic field $\varepsilon_n(\mathbf{k})$ is used to define the closed semiclassical orbits and the corresponding area $S_0(\epsilon)$, the modified semiclassical approach employs the modified band structure given by

$$E_{n,B}(\mathbf{k}) = \varepsilon_n(\mathbf{k}) + \mu_n(\mathbf{k})B, \quad (3)$$

where $\mathbf{B} = B\hat{z}$ is the magnetic field, $n$ is the band index, and $\mu_n(\mathbf{k})$ is the orbital magnetic moment of the $n$th magnetic band in the $z$-direction arising from the self-rotation of the corresponding wave packet[8]. The explicit form of $\mu_n(\mathbf{k})$ is

$$\mu_n(\mathbf{k}) = \frac{e}{\hbar}\,\mathrm{Im}\langle\partial_x u_n(\mathbf{k})|[\varepsilon_n(\mathbf{k}) - H(\mathbf{k})]|\partial_y u_n(\mathbf{k})\rangle, \quad (4)$$

where $H(\mathbf{k})$ is the Hamiltonian in momentum space and $\partial_i = \partial_{k_i}(i = x, y)$. Hence, the second term on the right-hand side of Eq. (3) indicates the leading energy correction from the orbital magnetic moment coupled to the magnetic field. In usual dispersive bands, the $B$-linear quantum correction is negligibly small in weak magnetic field limit compared to the zero-field bandwidth. This is the reason why the original Onsager's semiclassical scheme in Eq. (1) works well.

In the case of a flat band with zero bandwidth, on the other hand, the $B$-linear quantum correction always dominates the modified band structure $E_{n,B}(\mathbf{k})$ in Eq. (3) even in a weak magnetic field limit. Moreover, the modified band dispersion of an IFB is generally dispersive so that the relevant semiclassical orbits can be defined unambiguously. As a result, one can obtain the Landau levels of the IFB in the adjacent gapped regions by applying the semiclassical quantization rule to $E_{n,B}(\mathbf{k})$, which naturally explains the LLS of the IFB. Especially, around the band

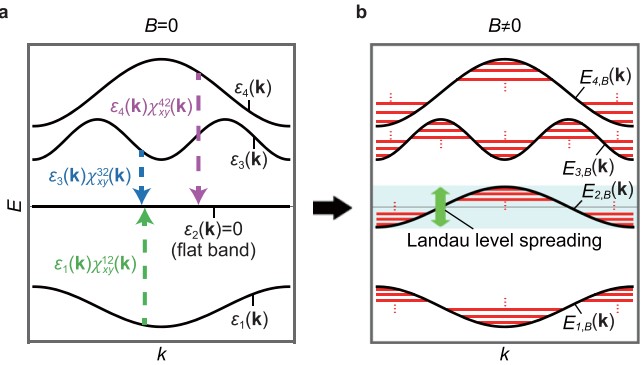

**Fig. 1 Landau level spreading (LLS) of an isolated flat band (IFB). a** The band structure of a two-dimensional system in the absence of a magnetic field. The second band with the energy $\varepsilon_2(\mathbf{k}) = 0$ corresponds to the IFB. The inter-band coupling $\varepsilon_m(\mathbf{k})\chi_{xy}^{m2}(\mathbf{k})$ of the IFB with the other dispersive band of the energy $\varepsilon_m(\mathbf{k})(m = 1, 3, 4)$ is indicated by a dashed vertical arrow. **b** The modified band dispersion $E_{n,B}(\mathbf{k})(n = 1, ..., 4)$ in the presence of the magnetic field. The corresponding Landau levels are shown by red solid lines. The LLS of the IFB is represented by the green arrow.

edges of $E_{n,B}(\mathbf{k})$, one can define the effective mass $m^*$, which is inversely proportional to $B$, from which the Onsager's scheme predicts Landau levels with a spacing $heB/m^* \propto B^2$. The resulting Landau spectrum is bounded by the upper and lower band edges of $E_{n,B}(\mathbf{k})$. The total magnitude $\Delta$ of the LLS is determined by the difference between the maximum and the minimum values of $E_{n,B}(\mathbf{k})$, namely, $\Delta = \max E_{n,B}(\mathbf{k}) - \min E_{n,B}(\mathbf{k})$. This result is valid as long as the band gap $E_{\text{gap}}$ between the IFB and its neighboring band at zero magnetic field is large enough, i.e., $E_{\text{gap}} \gg \max |E_{n,B}(\mathbf{k})|$. The generic behavior of an IFB under magnetic field is schematically described in Fig. 1 where one can clearly observe that the Landau levels of the IFB spread into the gaps at zero-field above and below the IFB.

**Geometric interpretation of the LLS.** Interestingly, we find that the LLS of IFBs is a manifestation of the non-trivial wave-function geometry of the flat band arising from inter-band couplings. One can show that the modified band dispersion of the IFB is given by

$$E_{n,B}(\mathbf{k}) = -2\pi \frac{\phi}{\phi_0} \frac{1}{A_0} \text{Im} \sum_{m \neq n} \varepsilon_m(\mathbf{k})\chi_{xy}^{nm}(\mathbf{k}), \quad (5)$$

in which

$$\chi_{ij}^{nm}(\mathbf{k}) = \langle \partial_i u_n(\mathbf{k})|u_m(\mathbf{k})\rangle \langle u_m(\mathbf{k})|\partial_j u_n(\mathbf{k})\rangle = A_i^{nm}(\mathbf{k})^* A_j^{nm}(\mathbf{k}). \quad (6)$$

where $\phi_0 = h/e$, $\phi = BA_0$ is the magnetic flux per unit cell, and $A_0$ is the unit cell area assumed to be $A_0 = 1$. Here, we assume that the $n$th band is the IFB at the zero-energy without loss of generality so that $\varepsilon_m(\mathbf{k})$ in Eq. (5) should be interpreted as the energy of the $m$th band with respect to the flat band energy. We note that $A_i^{nm}(\mathbf{k}) = \langle u_m(\mathbf{k})|\partial_i u_n(\mathbf{k})\rangle$ indicates the cross-gap Berry connection between the $n$th and $m$th bands ($n \neq m$) defined above, and $\chi_{ij}^{nm}(\mathbf{k})$ is the corresponding fidelity tensor that describes the transition amplitude between the $n$th and $m$th bands as discussed below. See Supplementary Notes 1 for the detailed derivation of Eq. (5). Hence, Eq. (5) indicates that the modified band dispersion of the IFB is given by the summation of the transition amplitudes $\chi_{xy}^{nm}(\mathbf{k})$ between the IFB and the $m$th band weighted by the energy $\varepsilon_m(\mathbf{k})$ of the $m$th band as illustrated in Fig. 1. This means that the immobile carriers with infinite effective mass in an IFB can respond to external magnetic field through the inter-

band coupling, characterized by the cross-gap Berry connection, to dispersive bands. The geometric character of the LLS is evident in our interpretation based on Eq. (5).

Let us discuss the geometric character of the fidelity tensor $\chi_{xy}^{nm}(\mathbf{k})$ more explicitly. In general, the geometry of the quantum state $u_n(\mathbf{k})$ can be derived from the Hilbert–Schmidt quantum distance[21–23] defined as

$$s(u_n(\mathbf{k}), u_n(\mathbf{k}')) = 1 - |\langle u_n(\mathbf{k})|u_n(\mathbf{k}')\rangle|^2, \quad (7)$$

which measures the similarity between $u_n(\mathbf{k})$ and $u_n(\mathbf{k}')$. For $\mathbf{k}' = \mathbf{k} + d\mathbf{k}$, we obtain

$$s(u_n(\mathbf{k}), u_n(\mathbf{k} + d\mathbf{k})) = \mathfrak{G}_{ij}^n(\mathbf{k})dk_i dk_j, \quad (8)$$

where $\mathfrak{G}_{ij}^n(\mathbf{k})$ indicates the quantum geometric tensor[24–26] whose explicit form is

$$\mathfrak{G}_{ij}^n(\mathbf{k}) = \sum_{m \neq n} \langle \partial_i u_n(\mathbf{k})|u_m(\mathbf{k})\rangle \langle u_m(\mathbf{k})|\partial_j u_n(\mathbf{k})\rangle = \sum_{m \neq n} \chi_{ij}^{nm}(\mathbf{k}), \quad (9)$$

which shows that the quantum geometric tensor $\mathfrak{G}_{ij}^n(\mathbf{k})$ of the $n$th band is given by the summation of the fidelity tensor $\chi_{ij}^{nm}(\mathbf{k})$ over all $m \neq n$. We note that $\chi_{ij}^{nm}(\mathbf{k})$ itself cannot define a distance as the triangle inequality is not satisfied. However, it is related to the transition probability or the fidelity $F(u_n(\mathbf{k}), u_m(\mathbf{k}'))$ between the $n$th and $m$th bands[27] through the following relations:

$$F(u_n(\mathbf{k}), u_m(\mathbf{k}')) = |\langle u_n(\mathbf{k})|u_m(\mathbf{k}')\rangle|^2, \quad (10)$$

$$F(u_n(\mathbf{k}), u_m(\mathbf{k} + d\mathbf{k})) = \chi_{ij}^{nm}(\mathbf{k})dk_i dk_j. \quad (11)$$

Thus, the geometric interpretation based on Eqs. (5) and (11) clearly show that the LLS originates from the inter-band coupling.

**Symmetry constraints on the LLS.** The LLS of an IFB is strongly constrained by symmetry. First, we consider a generic symmetry $\sigma$ whose action on the Hamiltonian is given by

$$U_\sigma(\mathbf{k})\overline{H(\mathbf{k})}^s U_\sigma(\mathbf{k})^\dagger = pH(O_\sigma\mathbf{k}), \quad (12)$$

where $s \in \{0, 1\}$, $p \in \{-1, 1\}$, and $U_\sigma(\mathbf{k})$ and $O_\sigma$ are unitary and orthogonal matrices representing $\sigma$, respectively. $\overline{x}^{s=1}$ denotes the complex conjugation of $x$ while $\overline{x}^{s=0} = x$. Note that $s = 0$ and 1 are relevant to the unitary and anti-unitary symmetries, respectively, while $p = -1$ and $+1$ correspond to anti-symmetry and symmetry, respectively.

Among all possible symmetries of the form in Eq. (12), we find that the modified band dispersion $E_{n,B}(\mathbf{k})$ vanishes when the system respects the chiral $C$ or space–time-inversion $I_{ST}$ symmetries in the zero magnetic flux (see the "Methods" section and Supplementary Notes 2 and 3 for the detailed derivation). $C$ and $I_{ST}$ are characterized by $(O_\sigma, s, p) = (\mathbb{1}, 0, -1)$ and $(\mathbb{1}, 1, 1)$, respectively, where $\mathbb{1}$ is the identity matrix. In the following, we demonstrate that the LLS is proportional to $B^2$ for a flat-band system with $I_{ST}$ symmetry in the zero magnetic fields, while the LLS is forbidden in the presence of chiral symmetry. Interestingly, although $I_{ST}$ symmetry would be broken as the magnetic field is turned on, the LLS is strongly constrained by $I_{ST}$ symmetry.

We further find that $\max E_{n,B}(\mathbf{k}) = -\min E_{n,B}(\mathbf{k})$ when the system respects a symmetry satisfying $(-1)^s p \text{ Det} O_\sigma = -1$ and $O_\sigma \neq \mathbb{1}$, such as time-reversal $T$ or reflection $R$ symmetry, at the zero magnetic field (see the "Methods" section and Supplementary Note 2 for detailed derivations). This implies that the minimum and maximum values of the LLS have the same magnitude but with opposite signs. The relevant tight-binding models are shown in Supplementary Notes 4.

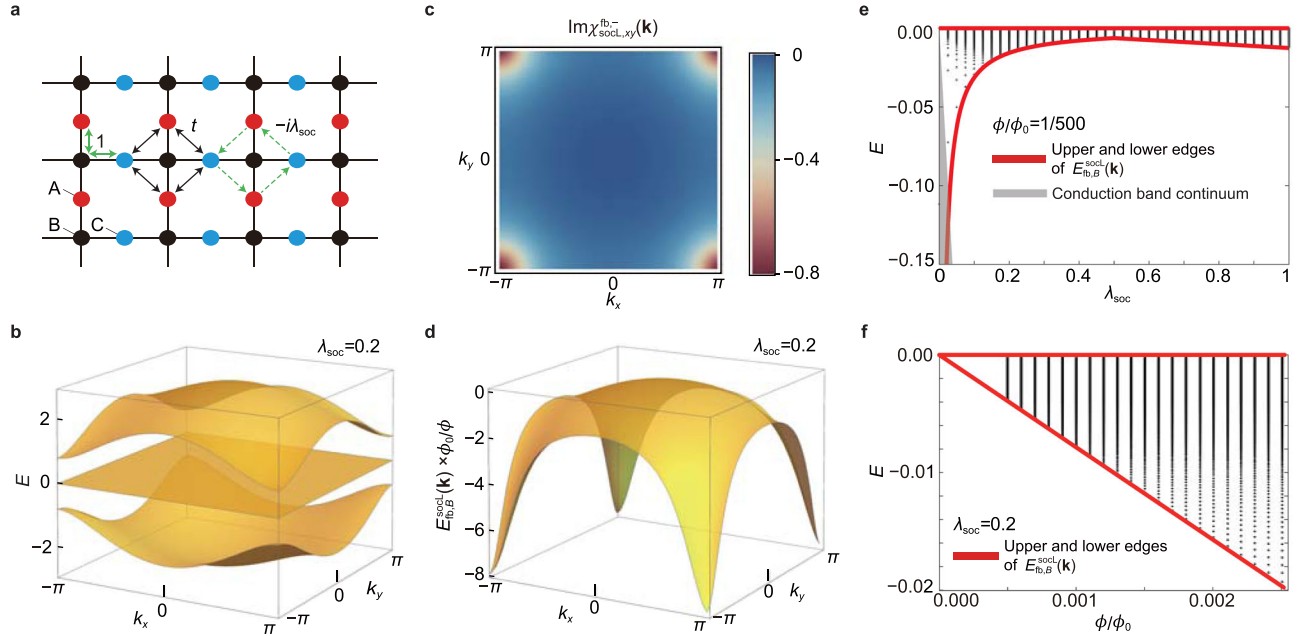

**Fig. 2 Landau level spreading of a generic system with an IFB. a** Lattice structure for the spin–orbit-coupled (SOC) Lieb model composed of three sublattices, A, B and C. The double-headed green arrows denote the nearest neighbor hoppings, and the single-headed green arrows indicate the spin–orbit coupling between A and C sublattices. The next-nearest hoppings $t$ between A and C is set to be zero ($t = 0.0$) in the SOC Lieb model. **b** The band structure of $H_{\text{socL}}(\mathbf{k})$ with $\lambda_{\text{soc}} = 0.2$. **c** Distribution of $\text{Im}\,\chi^{\text{fb},-}_{\text{socL},xy}(\mathbf{k})$. Note that $\text{Im}\,\chi^{\text{fb},-}_{\text{socL},xy}(\mathbf{k}) = -\text{Im}\,\chi^{\text{fb},+}_{\text{socL},xy}(\mathbf{k})$. **d** The modified band dispersion $E^{\text{socL}}_{\text{fb},B}(\mathbf{k})$ of the flat band in the presence of magnetic flux. **e** Landau level spectra of the flat band (black dots) as a function of $\lambda_{\text{soc}}$ for magnetic flux $\phi/\phi_0 = 1/500$. **f** Landau level spectra of the flat band (black dots) as a function of magnetic flux $\phi/\phi_0$ for $\lambda_{\text{soc}} = 0.2$.

**Generic flat-band systems**. We first consider the spin–orbit-coupled (SOC) Lieb model[28] as an example of generic flat-band systems. The lattice structure for this model is shown in Fig. 2a. The model consists of the nearest-neighbor hopping with the amplitude 1 and the spin–orbit coupling between the next nearest neighbor sites, which are denoted as green solid and dashed arrows, respectively, in Fig. 2a. The tight-binding Hamiltonian in momentum space is given by

$$H_{\text{socL}}(\mathbf{k}) = \begin{pmatrix} 0 & 2\cos\frac{k_y}{2} & -4i\lambda_{\text{soc}}\sin\frac{k_x}{2}\sin\frac{k_y}{2} \\ 2\cos\frac{k_y}{2} & 0 & 2\cos\frac{k_x}{2} \\ 4i\lambda_{\text{soc}}\sin\frac{k_x}{2}\sin\frac{k_y}{2} & 2\cos\frac{k_x}{2} & 0 \end{pmatrix},$$

(13)

where $\lambda_{\text{soc}}$ denotes the strength of spin–orbit coupling. The flat band's energy is zero, i.e., $\varepsilon_{\text{socL},\text{fb}}(\mathbf{k}) = 0$, and the energies of the other two bands are

$$\varepsilon_{\text{socL},\pm}(\mathbf{k}) = \pm 2\sqrt{\cos^2\frac{k_x}{2} + \cos^2\frac{k_y}{2} + 4\lambda^2_{\text{soc}}\sin^2\frac{k_x}{2}\sin^2\frac{k_y}{2}}, \quad (14)$$

which are plotted in Fig. 2b for $\lambda_{\text{soc}} = 0.2$. The band gap between the IFB and its neighboring bands is given by $4|\lambda_{\text{soc}}|$ if $|\lambda_{\text{soc}}| < 1/2$, and 2 if $|\lambda_{\text{soc}}| \geq 1/2$, thus the flat band is decoupled from other bands for non-zero $\lambda_{\text{soc}}$.

The analytic form of the fidelity tensor $\chi^{nm}_{xy}(\mathbf{k})$ is given by

$$\chi^{\text{fb},+}_{\text{socL},xy}(\mathbf{k}) = \left(\chi^{\text{fb},-}_{\text{socL},xy}(\mathbf{k})\right)^* = \frac{f(k_x,k_y)f(k_y,k_x)}{\left(\varepsilon_{\text{socL},+}(\mathbf{k})\right)^4(2 + \cos k_x + \cos k_y)},$$

(15)

where

$$f(k_x,k_y) = 4\lambda_{\text{soc}}\sin\frac{k_x}{2}\cos\frac{k_y}{2}(\cos^2\frac{k_x}{2} + 1) + i\varepsilon_{\text{socL},+}(\mathbf{k})\cos\frac{k_x}{2}\sin\frac{k_y}{2}.$$

Then, from Eq. (5), the modified band dispersion for the flat band is

given by

$$E^{\text{socL}}_{\text{fb},B}(\mathbf{k}) = -\frac{2\pi\lambda_{\text{soc}}\left(3 - \cos k_x - \cos k_y - \cos k_x \cos k_y\right)}{\left(\varepsilon_{\text{socL},+}(\mathbf{k})\right)^2}\frac{\phi}{\phi_0}.$$

(16)

In Fig. 2c, d, $\text{Im}\,\chi^{\text{fb},-}_{\text{socL},xy}(\mathbf{k})$ and $E^{\text{socL}}_{\text{fb},B}(\mathbf{k})$ are shown. We note that

$$\max E^{\text{socL}}_{\text{fb},B}(\mathbf{k}) = E^{\text{socL}}_{\text{fb},B}(0,0) = 0, \quad (17)$$

$$\min E^{\text{socL}}_{\text{fb},B}(\mathbf{k}) = \begin{cases} E^{\text{socL}}_{\text{fb},B}(\pi,\pi) = -\frac{\pi}{2\lambda_{\text{soc}}}\frac{\phi}{\phi_0} & (\lambda_{\text{soc}} \leq \frac{1}{2}) \\ E^{\text{socL}}_{\text{fb},B}(0,\pi) = -2\pi\lambda_{\text{soc}}\frac{\phi}{\phi_0} & (\lambda_{\text{soc}} > \frac{1}{2}) \end{cases}. \quad (18)$$

These minimum and maximum values of $E^{\text{socL}}_{\text{fb},B}(\mathbf{k})$ correspond to the lower and upper bounds for the LLS of the IFB as illustrated by red lines in Fig. 2e, f. Interestingly, the fidelity tensors $\chi^{\text{fb},+}_{\text{socL},xy}(\mathbf{k})$ and $\chi^{\text{fb},-}_{\text{socL},xy}(\mathbf{k})$ are conjugate of each other. This originates from the anti-unitary symmetry $C \cdot I_{\text{ST}}$, a combination of chiral $C$ and space–time-inversion $I_{\text{ST}}$ symmetries, present in the system (see Supplementary Note 2 for the details.)

**Chiral-symmetric system**. We construct a chiral-symmetric Lieb (c-Lieb) model as a representative example for chiral-symmetric IFB systems. The c-Lieb is defined on the same Lieb lattice as the SOC-Lieb model, but with different hoppings. As shown in Fig. 3a, this model consists only of the nearest-neighbor hoppings, denoted by green arrows. The hopping parameter from a B-site to a C-site is $t_1$ for the rightward hopping, and 1 for the leftward hopping. On the other hand, the hopping parameter from a B-site to an A-site is $t_2$ for the upward hopping, and 1 for the downward hopping. The corresponding tight-binding

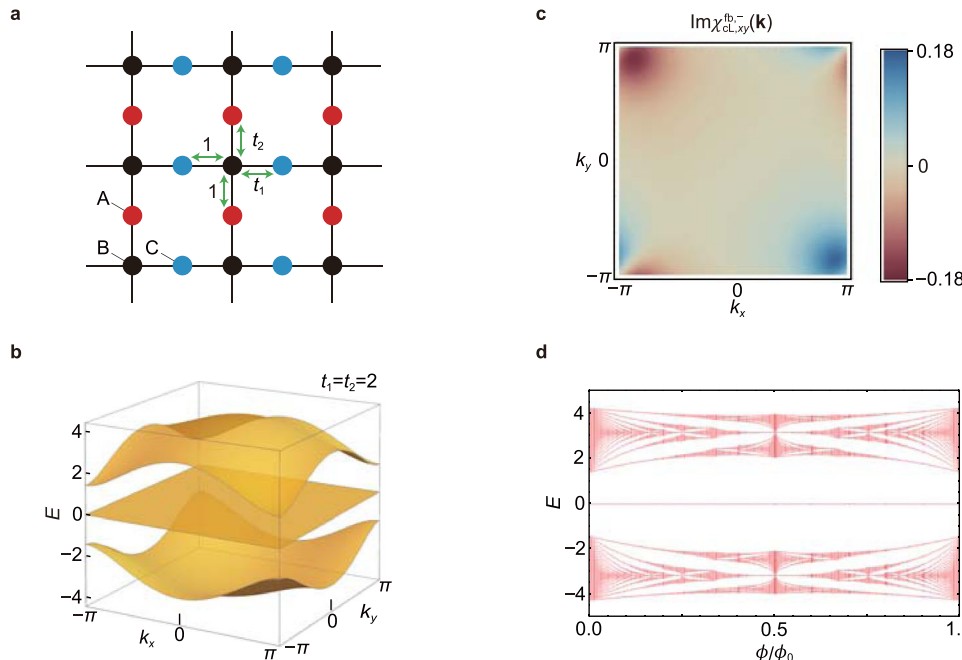

**Fig. 3 Landau level spreading of a flat-band system with chiral symmetry. a** Lattice structure for the c-Lieb model. The green arrows denote the hoppings. Note that the direction dependent hopping parameters induce a finite gap between the flat band and other bands. **b** The band structure for $H_{cL}(\mathbf{k})$ with $t_1 = t_2 = 2$. **c** Distribution of Im $\chi_{cL,xy}^{fb,-}(\mathbf{k})$. Due to the chiral symmetry, Im $\chi_{cL,xy}^{fb,-}(\mathbf{k}) =$ Im $\chi_{cL,xy}^{fb,+}(\mathbf{k})$ holds. **d** The Hofstadter spectrum of the c-Lieb model $H_{cL}(\mathbf{k})$ with $t_1 = t_2 = 2$. For any value of magnetic flux, the Landau levels of the flat band are fixed to the zero energy.

Hamiltonian in momentum space is given by

$$
H_{cL}(\mathbf{k}) = \begin{pmatrix} 0 & e^{i\frac{k_y}{2}} + t_2 e^{-i\frac{k_y}{2}} & 0 \\ e^{-i\frac{k_y}{2}} + t_2 e^{i\frac{k_y}{2}} & 0 & e^{-i\frac{k_x}{2}} + t_1 e^{i\frac{k_x}{2}} \\ 0 & e^{i\frac{k_x}{2}} + t_1 e^{-i\frac{k_x}{2}} & 0 \end{pmatrix}, \quad (19)
$$

with energy eigenvalues $\varepsilon_{cL,fb}(\mathbf{k}) = 0$ and $\varepsilon_{cL,\pm}(\mathbf{k}) = \pm\sqrt{2 + t_1^2 + t_2^2 + 2t_1\cos k_x + 2t_2\cos k_y}$. The chiral symmetry operator $C$ is given by $C = \text{Diag}(1, -1, 1)$ which gives a symmetry relation,

$$
CH_{cL}(\mathbf{k})C^{-1} = -H_{cL}(\mathbf{k}). \quad (20)
$$

Note that the wave function of the flat band is also a simultaneous eigenstate of the chiral symmetry having a definite chiral charge $c = +1$:

$$
C|u_{cL,fb}(\mathbf{k})\rangle = c|u_{cL,fb}(\mathbf{k})\rangle. \quad (21)
$$

Also, we obtain the fidelity tensor $\chi_{xy}^{fb,\pm}$, expressed by

$$
\chi_{cL,xy}^{fb,+}(\mathbf{k}) = \chi_{cL,xy}^{fb,-}(\mathbf{k})
$$
$$
= -\frac{(1 + t_1 e^{ik_x})(1 - t_1 e^{-ik_x})(1 + t_2 e^{-ik_y})(1 - t_2 e^{ik_y})}{8\left(\varepsilon_{cL,+}(\mathbf{k})\right)^4}. \quad (22)
$$

The band structure and Im $\chi_{cL,xy}^{fb,-}(\mathbf{k})$ are shown in Fig. 3b, c. Equation (22) indicates that the modified band dispersion $E_{fb,B}(\mathbf{k})$ vanishes for all $\mathbf{k}$ because $\varepsilon_{cL,+}(\mathbf{k}) = -\varepsilon_{cL,-}(\mathbf{k})$, which means that there is no LLS in the weak magnetic field. Also, we calculate the Hofstadter spectrum[29] for the c-Lieb model. Interestingly, we find that the LLS is absent even in the strong magnetic field, as shown in Fig. 3d. The existence of such zero-energy flat bands in the finite magnetic flux is guaranteed by chiral symmetry $C$. As explained in Supplementary Note 3, the minimal number of zero-energy flat bands is given by $|\text{Tr}[C]|$ at the zero magnetic flux.

Moreover, when the system has the $|\text{Tr}[C]|(>0)$ number of zero-energy flat bands at the zero magnetic flux, the LLS of the flat band(s) is forbidden unless a gap closes at zero energy $E = 0$ as the magnetic flux increases (see Supplementary Note 3). In the c-Lieb model, such a gap closing at $E = 0$ does not occur at any magnetic flux. Hence, there is no LLS in all range of magnetic flux. On the other hand, when a gap closes at $E = 0$ as the magnetic flux increases, the LLS is forbidden only in a finite range of magnetic flux. As an example, in Supplementary Note 4, we show the Hofstadter spectrum of the ten-band model for twisted-bilayer graphene proposed in ref. [30].

**Space–time-inversion-symmetric system.** The LLS of an IFB is weakly dependent on the magnetic field when the system respects space–time-inversion $I_{ST}$ symmetry at zero magnetic field. We consider spinless fermions on the checkerboard lattice shown in Fig. 4a, which is sometimes called the Tasaki or decorated square lattice[31–33]. This model respects both time-reversal $T$ and inversion $I$ symmetries. Hence, a combined symmetry, space–time-inversion symmetry $I_{ST} = I \bullet T$, exists. We note that the following discussion holds even if $T$ and $I$ are broken as long as $I_{ST}$ is not broken. The tight-binding Hamiltonian consists of the hopping processes up to the third nearest-neighbor hopping. In momentum space, the Hamiltonian is written as

$$
H_{I_{ST}}(\mathbf{k}) = \begin{pmatrix} 1 & \cos\frac{k_+}{2} + 2t\cos\frac{k_-}{2} \\ \cos\frac{k_+}{2} + 2t\cos\frac{k_-}{2} & 2t\cos k_x + 2t\cos k_y + \cos^2\frac{k_+}{2} + 4t^2\cos^2\frac{k_-}{2} \end{pmatrix}, \quad (23)
$$

$$
= \begin{pmatrix} 1 \\ \cos\frac{k_+}{2} + 2t\cos\frac{k_-}{2} \end{pmatrix} \begin{pmatrix} 1 & \cos\frac{k_+}{2} + 2t\cos\frac{k_-}{2} \end{pmatrix}, \quad (24)
$$

where $k_\pm = k_x \pm k_y$. For $t = 1.0$, the band structure is shown in Fig. 4b. This system hosts a flat band with zero energy and a dispersive band with positive energy. The energy eigenvalues are given by $\varepsilon_{I_{ST},\uparrow}(\mathbf{k}) = 1 + \left(\cos\frac{k_+}{2} + 2t\cos\frac{k_-}{2}\right)^2$ and $\varepsilon_{I_{ST},fb}(\mathbf{k}) = 0$.

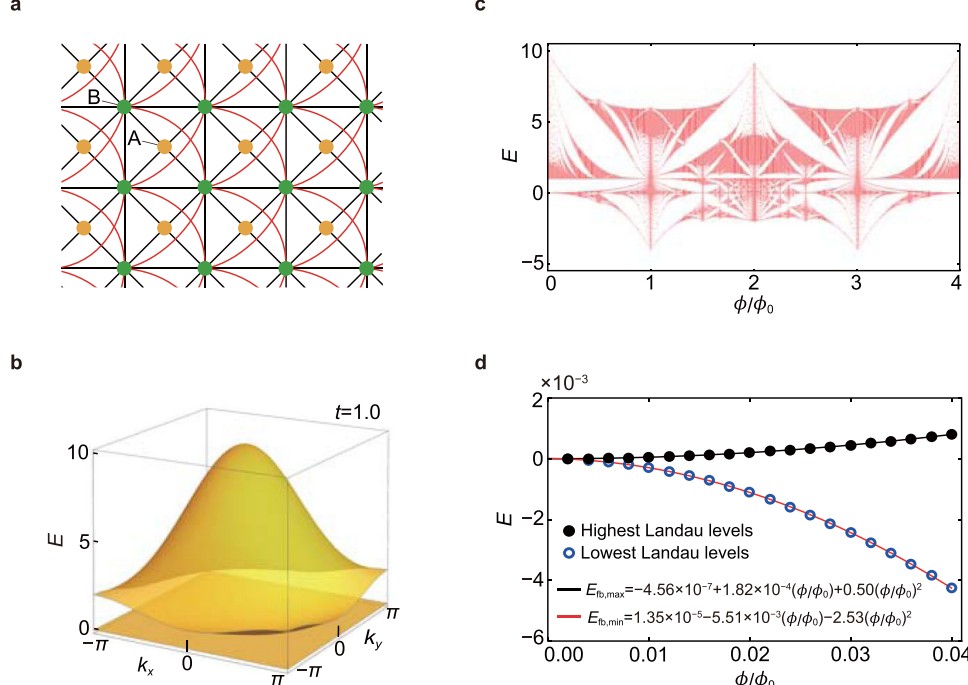

**Fig. 4 Landau level spreading of a flat-band system with space–time-inversion symmetry. a** The lattice structure for the $I_{ST}$-symmetric checkerboard model. The red and black lines denote the hopping processes between $A$ and $B$ sublattices. **b** The band structure of $H_{I_{ST}}(\mathbf{k})$ with $t = 1.0$. **c** The Hofstadter spectrum of the $I_{ST}$-symmetric checkerboard model with $t = 1.0$. **d** Variation of the lowest and highest Landau levels related to the flat band in the weak magnetic field (blue and black circles). $E_{fb,min}(\phi/\phi_0)$ and $E_{fb,max}(\phi/\phi_0)$ indicate the corresponding fitting functions (red and black solid lines) that exhibit quadratic magnetic field dependence dominantly.

In this system, $I_{ST}$ is simply given by the complex conjugation, i.e., $I_{ST} = \mathcal{K}$ and

$$I_{ST}|u_{I_{ST},fb/\uparrow}(\mathbf{k})\rangle = |u_{I_{ST},fb/\uparrow}(\mathbf{k})\rangle. \quad (25)$$

Also, explicit calculations show $\text{Im}\,\chi^{fb,\uparrow}_{I_{ST},xy}(\mathbf{k}) = 0$ and the vanishing modified band dispersion for the flat band $E_{n,B}(\mathbf{k}) = 0$, which results from Eq. (25). In Supplementary Note 2, we have proved that space–time inversion $I_{ST}$ imposes $E_{n,B}(\mathbf{k}) = 0$ in general. We also note that $E_{n,B}(\mathbf{k}) = 0$ is consistent with the fact that the orbital angular momentum, which is proportional to the orbital magnetic moment, is constrained to be zero in $I_{ST}$-symmetric systems. Although the LLS is negligible in the weak magnetic field, it becomes considerably large in the strong magnetic field as shown in the Hofstadter spectrum in Fig. 4c. As shown in Fig. 4c, the Landau levels of the flat band acquire or lose their energy as the magnetic flux increases from 0 to some finite value much less than 1. This implies that the higher-order corrections of the magnetic field must be considered. Although it is out of the scope of this work, we present a fitting of the highest and lowest Landau levels of the flat band with respect to the magnetic flux:

$$E_{fb,min}(\phi/\phi_0) = 1.35 \times 10^{-5} - 5.51 \times 10^{-3}(\phi/\phi_0) - 2.53(\phi/\phi_0)^2, \quad (26)$$

$$E_{fb,max}(\phi/\phi_0) = -4.56 \times 10^{-7} + 1.82 \times 10^{-4}(\phi/\phi_0) + 0.50(\phi/\phi_0)^2, \quad (27)$$

which is plotted in Fig. 4d where one can observe the dominant quadratic dependence on the magnetic field.

Finally, we comment on the gap closing at $(\phi/\phi_0, E) = (1, 1.0)$ in the Hofstadter spectrum in Fig. 4c. At $(\phi/\phi_0, E) = (1, 1.0)$, the Landau levels related to the flat and dispersive bands show a

closing of an indirect gap. We note that there is no closing of direct gaps in the Hofstadter Hamiltonian. Unlike the inevitable closing of the direct gap between topological bands in the finite magnetic flux reported before[11, 34], it is not necessary to close a direct gap in our system.

## Discussion

We have shown that the LLS of an IFB is determined by its wave-function geometry and the underlying symmetry of the system. The idea presented in this work goes beyond the conventional semiclassical idea in which the Landau level spectrum is dominantly determined by the band dispersion at zero magnetic fields. So far, we have focused on cases when the bandwidth of the IFB is strictly zero. However, in real materials, it is difficult to observe perfect flat bands due to the long-range hoppings and spin–orbit coupling[35–39]. To understand the influence of finite bandwidth of the IFB, we have studied another tight-binding model defined in the Lieb lattice including spin–orbit coupling. The hopping parameters, the band structure, and the LLS of this system are described in Figs. 2a, 5a–c, respectively. Under weak magnetic flux with $t\lambda_{soc}\phi > 0$, the LLS of the IFB cannot be observed because it is dominated by the energy scale of the bandwidth of the nearly flat band. However, the anomalous LLS arising from the wave-function geometry can be observed for the magnetic flux larger than a threshold value $(\phi/\phi_0)_{thres} \sim 8t\lambda_{soc}/\pi$ (see Fig. 5c). On the other hand, the LLS is not disturbed by the bandwidth when $t\lambda_{soc}\phi < 0$, because the nearly flat band has only positive energy (see Fig. 5c). Such a Lieb lattice model with spin–orbit coupling hosting a nearly flat band was already realized in an exciton-polariton system[40], and also is expected to be realized in electronic systems consisting of covalently bonded organic frameworks[41].

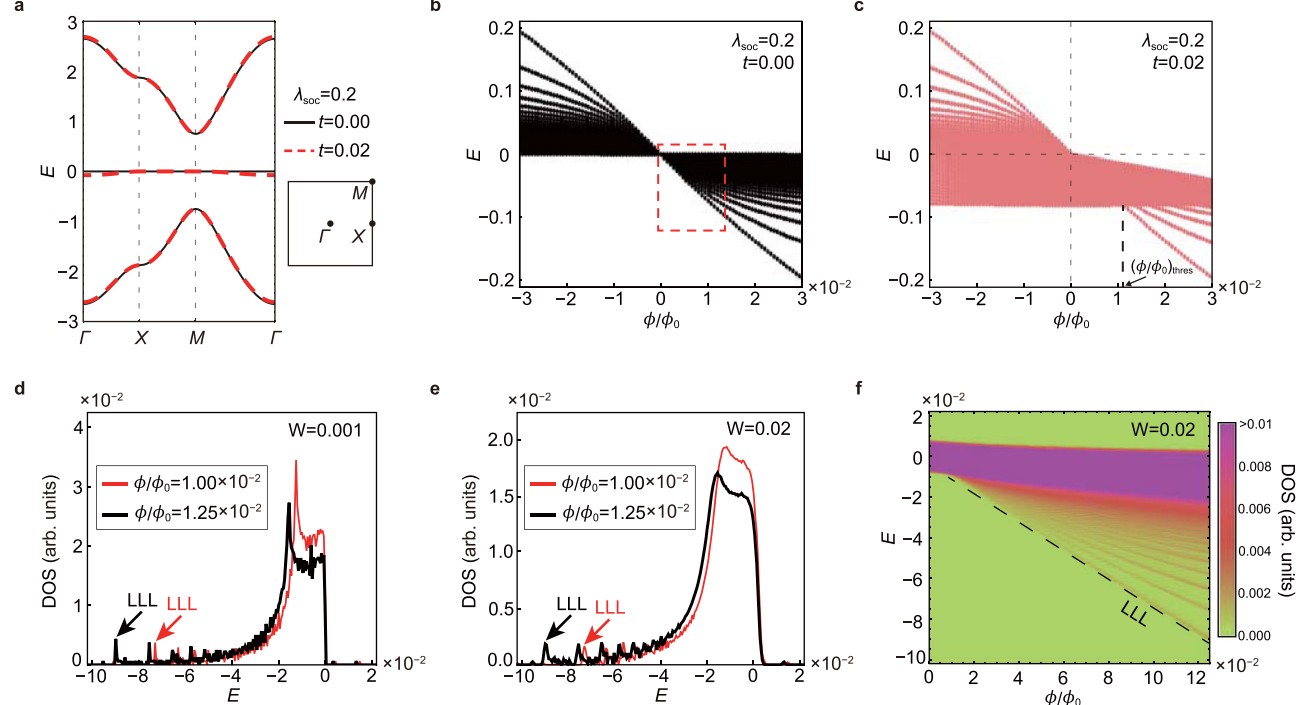

**Fig. 5 Influence of finite bandwidth and disorder on the Landau level spreading of IFBs. a–c** A nearly flat band in the spin-orbit coupled Lieb lattice where the dispersion of the FB arises from the next nearest neighbor hoppings between $A$ and $C$ sublattices with the amplitude $t$ (see Fig. 2a). **a** The band structure of the nearly-flat-Lieb model with $(t, \lambda_{soc}) = (0, 0.2)$ and $(0.02, 0.2)$, which are denoted by the black and red lines, respectively. The right side of the band structure describes the first Brillouin zone and the high-symmetry points. **b** The Landau levels in the weak magnetic field for $t = 0.00$ and $\lambda_{soc} = 0.2$. The lowest (highest) Landau level has the energy $-\frac{\pi}{2\lambda_{soc}} \phi/\phi_0$ when $\phi > 0$ ($\phi < 0$). **c** Similar plot for $t = 0.02$ and $\lambda_{soc} = 0.2$. When $\phi > 0$, the LLS can be observed once the magnetic flux exceeds the threshold value $(\phi/\phi_0)_{thres} \sim \frac{8t\lambda_{soc}}{\pi}$ because of the finite band width $4t$. While, the threshold magnetic flux is zero when $\phi < 0$ since the LLS develops only in the negative energy direction. **d–f** Density of states (DOS) of Landau levels in the presence of disorder. **d** DOS of Landau levels in very clean system with $W = 0.001$. The red (black) line corresponds to $\phi/\phi_0 = 1.00 \times 10^{-2}$ ($1.25 \times 10^{-2}$). The lowest Landau level (LLL) peaks are denoted by black and red arrows. The energy range of the plot corresponds to the red dashed box in (**b**). **e** DOS of Landau levels for $W = 0.02$. **f** The intensity plot of the DOS (or the Landau fan diagram) for $W = 0.02$ and the iteration number $N_{itr} = 20$ as a function of $\phi/\phi_0$. The plot ranges for the energy and magnetic flux correspond to the red dashed box in (**b**). The black dashed line indicates the LLL peak positions. Note that the maximum value of DOS in the plotted region is $2.306 \times 10^{-2}$ (arb. unit).

Finally, we discuss the influence of disorder on the LLS of an IFB and the related Landau level fan diagram. The fan diagram is obtained by calculating the density of states (DOS) of Landau levels of the disordered SOC Lieb model including a random impurity potential whose maximum strength is denoted by $W$ (see the "Methods" section for details). As shown in Fig. 5d, e, aside from the huge and wide DOS peaks from the dense Landau levels with higher Landau level indices, one can find small but sharp peaks corresponding to the LLLs of the IFB, from which the LLS of the IFB can be determined. While this LLL peak is buried in the DOS envelope of the higher Landau levels in the weak magnetic field, it splits away from this envelope as the magnetic field is large enough as shown in Fig. 5f. From the fan diagram, one can check the geometric principle described by Eq. (5) by extracting the slope of the LLL, which is represented by the dashed guideline in Fig. 5f. Here we considered the magnetic fluxes $\phi/\phi_0$ below $1.25 \times 10^{-2}$, which correspond to the experimentally accessible region. Note that when the size of a unit cell is equal to $l$nm, the relevant magnetic field is about $B \sim 4000 \times \phi/\phi_0 \times l^{-2}(T)$ approximately. For instance, in the case of the Lieb lattice composed of the covalently bonded organic frameworks[41], $\phi/\phi_0 = 1.25 \times 10^{-2}$ corresponds to $B \sim 50$ T. We expect the DOS peak corresponding to the LLL to be detected by the resistance measurement from magnetotransport experiments or the d$I$/d$V$ measurement from the scanning tunneling spectroscopy if the magnetic field is strong enough or the system is

sufficiently clean so that the Landau level spacing becomes larger than the Landau level broadening. Especially, when the LLS develops asymmetrically, like in Fig. 5b, an overall energy shift of the DOS from the flat band's energy appears more prominently, which provides a direct experimental signature of the LLS even in disordered systems.

Up to now, our discussion has been focused on conventional materials to realize flat bands. However, it is worth noting that there are various artificial systems such as photonic systems[13, 42–44], optical lattices[45–50], and systems with synthetic dimensions[51–55], which could offer better opportunities to test our theoretical prediction. In these systems, band engineering is relatively easier, and controlled experiments with artificial magnetic fields can also be performed. Designing realistic experimental setups for observing LLS of flat bands in such artificial systems would be one important problem for future study.

## Methods

**Symmetry constraints on the LLS.** In order to derive the symmetry constraints on $E_{n,B}(\mathbf{k})$ and $\chi_{xy}^{nm}(\mathbf{k})$, let us consider a symmetry operation $\sigma$ acting on the Hamiltonian,

$$U_\sigma(\mathbf{k})\overline{H(\mathbf{k})}^s U_\sigma(\mathbf{k})^\dagger = pH(O_\sigma\mathbf{k}), \qquad (28)$$

where $s \in \{0, 1\}$, $p \in \{-1, 1\}$, $U_\sigma$ indicates a unitary matrix representing the symmetry $\sigma$, and $\bar{x} = x^*$ means the complex conjugation of $x$. From now on, we use a compact notation $\mathfrak{g}_\sigma = (O_\sigma, s, p)$ to describe the operation of the symmetry $\sigma$. For example, $\mathfrak{g}_T = (-\mathbb{1}_d, 1, 1)$ is used for time-reversal symmetry $T$ where $d$ and $\mathbb{1}_d$ denote the dimensionality and the $d \times d$ identity matrix, respectively. The

symmetry constraints on $E_{n,B}(\mathbf{k})$ and $\chi_{ij}^{nm}(\mathbf{k})$ derived from Eq. (28) are

$$E_{n,B}(O_\sigma \mathbf{k}) = (-1)^s p \ \mathrm{Det} O_\sigma \ E_{n,B}(\mathbf{k}), \tag{29}$$

$$\chi_{ij}^{nm_\sigma}(O_\sigma \mathbf{k}) = [O_\sigma]_{ii'}[O_\sigma]_{jj'}\overline{\chi_{i'j'}^{nm}(\mathbf{k})}^{\,s}, \tag{30}$$

where the band indices $m$ and $m_\sigma$ in Eq. (30) are chosen such that $\varepsilon_m(\mathbf{k}) = p \ \varepsilon_{m_\sigma}(O_\sigma \mathbf{k})$. Detailed derivation of Eqs. (29) and (30) and comments on the degenerate bands can be found in Supplementary Note 2. From equation (29), we obtain two symmetries that give vanishing modified band dispersion, $E_{n,B}(\mathbf{k}) = 0$: $\mathfrak{g}_C = (\mathbb{1}_d, 0, -1)$ and $\mathfrak{g}_{I_{ST}} = (\mathbb{1}_d, 1, 1)$ which correspond to chiral symmetry $C$ and space–time-inversion symmetry $I_{ST}$, respectively. On the other hand, when $(-1)^s \mathrm{Det} O_\sigma = -1$ and $\mathrm{Det} O_\sigma \neq \mathbb{1}$, the modified band dispersion satisfies $E_{n,B}(O_\sigma \mathbf{k}) = -E_{n,B}(\mathbf{k})$, which implies max $E_{n,B}(\mathbf{k}) = -$ min $E_{n,B}(\mathbf{k})$. Time-reversal $T$ and reflection $R$ symmetries belong to this case. Also, the contribution to the $E_{n,B}(\mathbf{k})$ from each band via the inter-band coupling in Eq. (5) can be systematically understood by using equation (30) (see Supplementary Note 2 for details).

**Calculation scheme for the Landau levels**. We calculate the Hofstadter spectrum by numerically implementing the Peierls substitution to the tight-binding Hamiltonian[29].

**Calculation of Landau fan diagram including disorder**. To obtain the Landau fan diagram including disorder effect, we study a finite-size SOC Lieb model $H_{\mathrm{socL}}(\mathbf{k})$ composed of 40 by 40 unit cells. Disorder is introduced by the Hamiltonian $H_{\mathrm{dis}}$ with components $(H_{\mathrm{dis}})_{ij} = w_i \delta_{ij}$, where $i, j = 1, \dots, 4800$ denotes the unit cell index and $w_i \in [-W/2, W/2]$ follows a uniform probability distribution. By diagonalizing the disordered Hamiltonian $N_{\mathrm{itr}} = 200$ times and averaging the results, the density of states (DOS) of Landau levels is obtained. Note that chiral edge states are found in the gap between flat and dispersive bands. It is because the two dispersive bands in the SOC Lieb model have the Chern number ±1, respectively depending on the sign of spin–orbit coupling, despite the topologically trivial middle flat band. However, the contribution of edge states to DOS is quantitatively negligible.

## Data availability

The data that support the findings of this study are available from the corresponding authors upon reasonable request.

## Code availability

The numerical codes used in this paper are available from the corresponding authors upon reasonable request.

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

## Acknowledgements

Y.H. was supported by IBS-R009-D1 and Samsung Science and Technology Foundation under Project Number SSTF-BA2002-06. J.-W.R. was supported by IBS-R009-D1, and the National Research Foundation of Korea (NRF) Grant funded by the Korea government (MSIT) (Grant No. 2021R1A2C1010572). B.-J.Y. was supported by the Institute for Basic Science in Korea (Grant No. IBS-R009-D1), Samsung Science and Technology Foundation under Project Number SSTF-BA2002-06, the National Research Foundation of Korea (NRF) Grant funded by the Korea government (MSIT) (No.2021R1A2C4002773, and No. NRF-2021R1A5A1032996), and the U.S. Army Research Office and Asian Office of Aerospace Research & Development (AOARD) under Grant No. W911NF-18-1-0137.

## Author contributions

Y.H. and J.-W.R. performed theoretical and numerical analyses. B.-J.Y. supervised the project. All authors analysed the data. The manuscript was written by all authors.

## Competing interests

The authors declare no competing interests.
