## [Peer Review File · Nature Communications]

Editorial Note: Parts of this peer review file have been redacted as indicated to remove third-party material where no permission to publish could be obtained.

Reviewers' comments:

Reviewer #1 (Remarks to the Author):

The authors have studied the impact of orbital magnetic fields on tight binding models with isolated flat bands. They find that the Landau levels of such flat bands have a finite spread, which they can relate to "geometrical" properties like the cross-gap Berry connection. They then show that symmetry considerations can also lead to predictions on the properties of Landau levels appearing from isolated flat bands.

While the paper is well written and scientifically sound, I think it would be better suited for a more specialized journal. The intricacies of Landau levels in lattice systems are a somewhat specialized topic which I don't think would be appealing to the broad audience of Nature Communications. Even within the specialized topic of Landau levels in lattice systems, I don't think the results are really surprising. As far as I can tell, there is no fundamental reason why an isolated flat band could not have a finite spread of its Landau levels, so I am not sure why the authors call this phenomenon anomalous. Further, the theoretical techniques and tools they use are standard in the field and do not constitute a novelty in themselves. The relation to potential experiments and materials is pretty remote, so I think this paper would be better suited for a more specialized journal.

I have one specific question: in the abstract, the authors mention that "Landau levels of a band are conventionally believed to be bounded by its upper and lower band edges at 12 zero magnetic field". Is that just a belief, or are there some existing proofs in certain cases? Any references to justify this claim?

I think the authors should also reference early work on Landau levels in lattices, like the seminal work of Hofstadter:
PhysRevB.14.2239

Reviewer #2 (Remarks to the Author):

In this work the authors studied possible Landau levels that can emerge out of isolated flat bands under applied magnetic field. This is an interesting question since it certainly requires one to go beyond Onsager's semiclassical argument. The question may also be relevant for some realistic systems.

The authors approached this problem essentially in a self-consistent manner: they pointed out that the flat band will acquire an effective dispersion once a magnetic field is applied, and then used Onsager's semiclassical result using the modified dispersion. The authors also pointed out a relation between the modified dispersion and the geometry of the Bloch wave functions (in zero field). Various constraints from symmetries are also studied.

My main concern about this approach is to what extent is Onsager's semiclassical result (Eq. (1)) applicable in the regime studied in this work. The standard semiclassical argument is reliable only if the magnetic field is weak. In the regime studied in this work, it seems that the field can never be considered "weak", so why should we trust Eq. (1)? Admittedly, this line of thinking does work even in the strong field regime in the most familiar cases of Schrodinger and Dirac fermions, but I'm not aware of any deep reason why it should work beyond the two special cases. If the authors do have an argument it will be helpful to explicitly spell it out.

The paper is otherwise quite well written, and will be potentially very interesting if the logic is solid. I'm open to recommend publishing it if the authors can address my concern to some degree.

Reviewer #3 (Remarks to the Author):

In "Geometric characterization of anomalous Landau levels of isolated flat bands" the authors propounded that Landau levels exist across a spread of energies, even from a single zero-dispersion band in the zero magnetic field limit. This spread is dependent on the non-abelian gauge field of the bands, and is most apparent when it is originally a flat band. The authors also presented a study of several models, both in the main text and supplement, with details delightfully worked out.

Overall, this work is very well written, interesting and pedagogical, and deserves publication in a journal of very good standing, for instance Communications Physics. However, I hesitate recommending publication in a top journal like Nature Communications for the following reasons:

1. In this work, the main novelty seems to be that flat bands that are not chiral symmetric exhibits significant spectral spread. However, part of the key theory i.e. Eq 1 and Eq 3 are already derived in Ref 8. What that seems new to some extent is Eq 5. However, that is not explicitly derived in this manuscript.
2. Even if not totally conceptually novel, the results will still have been of great significance if they corroborate well with experimental results. For instance, one may think that this spectral spread of Landau levels can be considered as a very experimentally salient probe of nontrivial non-abelian band geometry. However, the authors hardly mentioned specific experiments on their theoretical predictions.

Some additional comments:

- a) Eq 5 is the main result used in this work, and the authors should provide a full derivation of it, whether that can be found elsewhere or not. It is also interesting to see why it does not have any dependence on ϵ_n , even though the mathematically equivalent expression (Eq 3) depends explicitly on ϵ_n . In other words, how can $E_{\{n,B\}}(k)$ "know" about ϵ_n , which can be shifted up or down at will, independent of the other bands?
- b) In Fig 1 and the discussion, it is clear that the modified band structure $E_{\{n,B\}}(k)$ does not give the actual energy levels of the actual Landau levels, even though it determines the energy range of the Landau levels. Is this interpretation correct? If yes, the authors should make that clear. Also, what determines the spacing between the Landau levels within the energy spread ?

1 Reply to the first reviewer

The authors have studied the impact of orbital magnetic fields on tight binding models with isolated flat bands. They find that the Landau levels of such flat bands have a finite spread, which they can relate to “geometrical” properties like the cross-gap Berry connection. They then show that symmetry considerations can also lead to predictions on the properties of Landau levels appearing from isolated flat bands.

Author’s Response: We thank the reviewer for reading our manuscript carefully.

While the paper is well written and scientifically sound, I think it would be better suited for a more specialized journal. The intricacies of Landau levels in lattice systems are a somewhat specialized topic which I don’t think would be appealing to the broad audience of Nature Communications. Even within the specialized topic of Landau levels in lattice systems, I don’t think the results are really surprising. As far as I can tell, there is no fundamental reason why an isolated flat band could not have a finite spread of its Landau levels, so I am not sure why the authors call this phenomenon anomalous.

Author’s Response: We thank the reviewer for mentioning that our manuscript is scientifically sound and well-written. Regrettably, however, the reviewer concluded that the topic we studied does not appeal to the broad audience, and the spreading of Landau levels is not really surprising.

Respectfully, we disagree with the reviewer’s opinion that our paper lacks the general interest for the following reasons. First, our main point is not simply to show the intricacies of the Landau levels in certain lattices but to establish a new geometric principle for understanding the Landau level structure of general flat band systems. The analysis of Landau levels using the Onsager’s semiclassical theory has been one of the most important tools to extract the Berry phase, successfully applied to various band structures including conventional parabolic bands and relativistic Dirac dispersion. However, this powerful semiclassical method does not work in flat bands where semiclassical orbits are ill-defined. The Onsager’s semiclassical scheme predicts no Landau level formation of isolated flat bands generally, which is not consistent with quantum mechanical calculations in various flat band models. In this work, we have shown that there is a class of flat bands exhibiting anomalous Landau level spectra, which are *anomalous* as they are beyond the description of the Onsager’s semiclassical framework. Moreover, the anomalous Landau level spectra are governed by a geometric quantity of wave functions, called the cross-gap Berry connection. Our results establish a new geometric principle for characterizing the Landau levels of flat bands, which definitely goes beyond the conventional semiclassical theory based on Onsager’s quantization rule.

Second, the Bloch wavefunction’s geometry and flat band, which are two important keywords of our work, have received great attention in a wide range of research fields. The geometry of wave functions has been a central notion in vast physical phenomena ranging from the Aharonov-Bohm effect to the topological phases of matter. Also, the flat band research is one of the most rapidly growing research areas in various fields including solid state physics, optics, metamaterial research, and so on. The geometry of the flat band’s wave function is a topic in which these two exciting topics are entangled is definitely an important problem, which we believe appeals to the broad readership of Nature Communications.

As for the novelty and the general interest of our work, we also would like to note the positive comments from the other two reviewers: the second reviewer mentioned that “*This is an interesting question since it certainly requires one to go beyond Onsager’s semiclassical argument. The question may also be relevant*

for some realistic systems” and “The paper is otherwise quite well written, and will be potentially very interesting if the logic is solid”. Also, the third reviewer commented that “Overall, this work is very well written, interesting and pedagogical, and deserves publication in a journal of very good standing,...”.

Considering the reviewer’s comment, we have significantly improved the abstract and introduction of our manuscript so that the novelty of our results is clearly explained.

Further, the theoretical techniques and tools they use are standard in the field and do not constitute a novelty in themselves. The relation to potential experiments and materials is pretty remote, so I think this paper would be better suited for a more specialized journal.

Author’s Response: As the reviewer pointed out, we used the standard theoretical techniques and tools. Nevertheless, the results we obtained, especially the geometrical description of the Landau level spreading of flat bands are novel because it goes beyond semiclassical scheme, the prevailing method which has been applied to various solid state physics.

We thank the reviewer for mentioning the potential experiments related to our theory. To address this question, we additionally calculated Landau fan diagram and evolution of density of states including disorder effect, which can be directly compared to magnetotransport or STM measurements. These new results are displayed in Figure 5 and in the last paragraph of the discussion section in the revised manuscript. We believe that these additional results fill the gap between the theory and potential experiments.

I have one specific question: in the abstract, the authors mention that “Landau levels of a band are conventionally believed to be bounded by its upper and lower band edges at zero magnetic field”. Is that just a belief, or are there some existing proofs in certain cases? Any references to justify this claim?

Author’s Response: We thank the reviewer for reading our paper carefully. By the expression “conventionally believed”, we mean “the well-known results from conventional semiclassical approach”. According to the semiclassical scheme, which is the most utilized tool for describing the physics of many solid state phenomena, the Landau levels can appear only within the energy regions where the density of states is nonzero at zero magnetic field. This is because the semiclassical orbit which satisfies the Onsager’s semiclassical quantization condition is defined using the band structure at zero field. Any Landau level that appears outside the upper and lower band edges at zero magnetic field is anomalous according to the semiclassical scheme based on the Onsager’s quantization condition. One recent paper where the Onsager’s semiclassical scheme is clearly explained is [“Landau levels, response functions and magnetic oscillations from a generalized Onsager relation”. SciPost Phys. 4, 024 (2018)].

Interestingly, some examples which violate the prediction of this semiclassical scheme have been discovered recently, such as topological bands or singular flat bands described by singular Bloch wave functions. Especially, in the paper [Biao Lian, Fang Xie, and B. Andrei Bernevig, Phys. Rev. B 102, 041402(R) (2020)], the authors provided a pedagogical argument why one can have intriguing in-gap Landau levels unbounded by the original band edges in topological bands. Also in the paper [J.W.Rhim, K.Kim, and B.J.Yang, Nature 584, 59-63 (2020)], it is shown that singular flat bands with a quadratic band crossing exhibit anomalous Landau levels outside the bounds of semiclassical theory.

Considering the reviewer’s comment, we revised the opening part of the abstract in a way that the novelty of our finding is clarified more clearly, as follows. “According to the Onsager’s semiclassical quantization rule, Landau levels of a band are bounded by its upper and lower band edges at zero magnetic field. Thus no Landau level is expected to appear in the gapped regions of the original band structure

where semiclassical orbits are forbidden. However, there are two notable examples violating this expectation, including topological bands with quantized invariants and flat bands with singular band crossings, whose wave functions possess some singularities. Here we introduce a distinct class of flat band systems where anomalous Landau level spreading (LLS) appears outside the zero-field energy bounds, although the relevant wave function is nonsingular. In particular, we establish a general geometric principle governing the anomalous LLS of isolated flat bands, using the cross-gap Berry connection that measures the wave function geometry of multi-bands.”

I think the authors should also reference early work on Landau levels in lattices, like the seminal work of Hofstadter: PhysRevB.14.2239

Author’s Response: We thank the reviewer for suggesting a relevant reference, which is now added in the revised manuscript.

2 Reply to the second reviewer

In this work the authors studied possible Landau levels that can emerge out of isolated flat bands under applied magnetic field. This is an interesting question since it certainly requires one to go beyond Onsager's semiclassical argument. The question may also be relevant for some realistic systems.

The authors approached this problem essentially in a self-consistent manner: they pointed out that the flat band will acquire an effective dispersion once a magnetic field is applied, and then used Onsager's semiclassical result using the modified dispersion. The authors also pointed out a relation between the modified dispersion and the geometry of the Bloch wave functions (in zero field). Various constraints from symmetries are also studied.

Author's Response: We thank the reviewer for reading our manuscript carefully and appreciating the motivation of our work.

My main concern about this approach is to what extent is Onsager's semiclassical result (Eq. (1)) applicable in the regime studied in this work. The standard semiclassical argument is reliable only if the magnetic field is weak. In the regime studied in this work, it seems that the field can never be considered "weak", so why should we trust Eq. (1)? Admittedly, this line of thinking does work even in the strong field regime in the most familiar cases of Schrodinger and Dirac fermions, but I'm not aware of any deep reason why it should work beyond the two special cases. If the authors do have an argument it will be helpful to explicitly spell it out.

Author's Response: We thank the reviewer for this important comment. As the reviewer pointed out, the Onsager's semiclassical scheme is generally valid in weak field limit. The reason why we included two Hofstadter butterfly diagrams in Fig. 3 and Fig. 4 is to examine how the results from the Onsager's scheme, which is valid in weak field limit, evolves as the magnetic field increases. From these additional calculations, we observed the following interesting results.

First, in the chiral symmetric case shown in Fig. 3, the Onsager's scheme predicts the absence of the Landau level spreading of the flat band. Interestingly, this conclusion remains valid even in strong field limit as shown in the Hofstadter's diagram. This clearly demonstrates that in chiral symmetric flat bands, the results obtained from the Onsager's scheme hold even in the strong magnetic field regime.

On the other hand, in the case of space-time-inversion symmetric systems shown in Fig. 4, although the Onsager's scheme predicts no Landau level spreading of flat bands, the Hofstadter butterfly diagram shows finite Landau level spreading as magnetic field increases. This shows that it is necessary to include higher-order correction terms in magnetic field to improve the semiclassical theory.

To reflect the fact that the semiclassical theory is generally valid in weak field limit, in the revised manuscript, we clearly mentioned that "This quantization rule is generally valid in weak field limit" below Eq. (1).

Additionally, let us make a short remark on the reviewer's comment "*Admittedly, this line of thinking does work even in the strong field regime in the most familiar cases of Schrodinger and Dirac fermions, but I'm not aware of any deep reason why it should work beyond the two special cases.*" In the case of Schrodinger and Dirac fermions, the corresponding Hamiltonian is scale invariant. Namely, the Hamiltonian keeps the same form as the momentum is rescaled. Because of this, the results of the semiclassical theory are invalid independent of the strength of the magnetic field. On the other hand, as lattices models are not scale invariant in general, the results in weak field limit can be broken in strong field limit, unless

the system has special symmetries such as the chiral symmetry.

The paper is otherwise quite well written, and will be potentially very interesting if the logic is solid. I'm open to recommend publishing it if the authors can address my concern to some degree.

Author's Response: We indeed thank the reviewer for his/her positive evaluation of our work.

3 Reply to the third reviewer

In "Geometric characterization of anomalous Landau levels of isolated flat bands" the authors propounded that Landau levels exist across a spread of energies, even from a single zero-dispersion band in the zero magnetic field limit. This spread is dependent on the non-abelian gauge field of the bands, and is most apparent when it is originally a flat band. The authors also presented a study of several models, both in the main text and supplement, with details delightfully worked out.

Overall, this work is very well written, interesting and pedagogical, and deserves publication in a journal of very good standing, for instance Communications Physics. However, I hesitate recommending publication in a top journal like Nature Communications for the following reasons:

Author's Response: We thank the reviewer for his/her positive evaluation of our work, and providing us fruitful suggestions. Below please find our responses to all questions/comments from the reviewer, which, we hope, are convincing enough so that the reviewer can now recommend our work for publication in Nature Communications.

1. In this work, the main novelty seems to be that flat bands that are not chiral symmetric exhibits significant spectral spread. However, part of the key theory i.e. Eq 1 and Eq 3 are already derived in Ref 8. What that seems new to some extent is Eq 5. However, that is not explicitly derived in this manuscript.

Author's Response: We thank the reviewer for this constructive comment. Considering the reviewer's comment, we added the detailed derivation of Eq. 5 in the Supplementary Note 1. Below we present the detailed derivation for the convenience of the reviewer.

In this section, we derive our main result, namely Eq. (5) in the main text. To this end, we start from the orbital magnetic moment formula Eq. (4), which is given by

$$\mu_n(\mathbf{k}) = \frac{e}{\hbar} \text{Im} \langle \partial_x u_n(\mathbf{k}) | [\varepsilon_n(\mathbf{k}) - H(\mathbf{k})] | \partial_y u_n(\mathbf{k}) \rangle. \quad (1)$$

While we are interested in the Landau level spreading of the flat band, let us assume that, in the band structure, the flat band is the n -th band with energy ε_0 . Then, Eq. (1) becomes

$$\mu_n(\mathbf{k}) = \frac{e}{\hbar} \text{Im} \langle \partial_x u_n(\mathbf{k}) | [\varepsilon_0 - H(\mathbf{k})] | \partial_y u_n(\mathbf{k}) \rangle, \quad (2)$$

$$= \frac{e}{\hbar} \text{Im} \langle \partial_x u_n(\mathbf{k}) | \sum_l |u_l(\mathbf{k})\rangle \langle u_l(\mathbf{k})| [\varepsilon_0 - H(\mathbf{k})] \sum_m |u_m(\mathbf{k})\rangle \langle u_m(\mathbf{k})| \partial_y u_n(\mathbf{k}) \rangle, \quad (3)$$

$$= \frac{e}{\hbar} \text{Im} \sum_{l,m} [\varepsilon_0 - \varepsilon_m(\mathbf{k})] \langle \partial_x u_n(\mathbf{k}) | u_l(\mathbf{k}) \rangle \langle u_m(\mathbf{k}) | \partial_y u_n(\mathbf{k}) \rangle \delta_{lm}, \quad (4)$$

$$= \frac{e}{\hbar} \text{Im} \sum_m [\varepsilon_0 - \varepsilon_m(\mathbf{k})] \langle \partial_x u_n(\mathbf{k}) | u_m(\mathbf{k}) \rangle \langle u_m(\mathbf{k}) | \partial_y u_n(\mathbf{k}) \rangle, \quad (5)$$

where the completeness relation $I = \sum_m |u_m(\mathbf{k})\rangle \langle u_m(\mathbf{k})|$ is used. Without loss of generality, one can assume that $\varepsilon_0 = 0$. Then, by inserting Eq. (5) into the energy formula in the main text, namely Eq. (3),

we obtain

$$E_{n,B}(\mathbf{k}) = \varepsilon_n(\mathbf{k}) + \mu_n(\mathbf{k})B = \mu_n(\mathbf{k})B, \quad (6)$$

$$= -B \frac{e}{\hbar} \text{Im} \sum_m \varepsilon_m(\mathbf{k}) \langle \partial_x u_n(\mathbf{k}) | u_m(\mathbf{k}) \rangle \langle u_m(\mathbf{k}) | \partial_y u_n(\mathbf{k}) \rangle, \quad (7)$$

$$= -2\pi \frac{\phi}{\phi_0} \text{Im} \sum_m \varepsilon_m(\mathbf{k}) \langle \partial_x u_n(\mathbf{k}) | u_m(\mathbf{k}) \rangle \langle u_m(\mathbf{k}) | \partial_y u_n(\mathbf{k}) \rangle, \quad (8)$$

where A_0 is the area of the unit cell, $\phi = A_0 B$, and $\phi_0 = h/e$. Finally, noting that the fidelity tensor is defined by

$$\chi_{ij}^{nm}(\mathbf{k}) = \langle \partial_i u_n(\mathbf{k}) | u_m(\mathbf{k}) \rangle \langle u_m(\mathbf{k}) | \partial_j u_n(\mathbf{k}) \rangle = A_i^{nm}(\mathbf{k})^* A_j^{nm}(\mathbf{k}), \quad (9)$$

we obtain

$$E_{n,B}(\mathbf{k}) = -2\pi \frac{\phi}{\phi_0} \frac{1}{A_0} \text{Im} \sum_{m \neq n} \varepsilon_m(\mathbf{k}) \chi_{xy}^{nm}(\mathbf{k}). \quad (10)$$

Since we assume that the flat band's energy is zero, one should interpret $\varepsilon_m(\mathbf{k})$ in Eq. (10) as the energy of the m -th band with respect to the flat band energy.

2. Even if not totally conceptually novel, the results will still have been of great significance if they corroborate well with experimental results. For instance, one may think that this spectral spread of Landau levels can be considered as a very experimentally salient probe of nontrivial non-abelian band geometry. However, the authors hardly mentioned specific experiments on their theoretical predictions.

Author's Response: We indeed thank the reviewer for this constructive comment. As the reviewer mentioned, it is important to make experimentally measurable predictions. Following the reviewer's comment, we newly computed the Landau level fan diagram of the disordered flat band model, taking the Lieb lattice model with the spin-orbit coupling as an example. The Landau level fan diagram is a plot of the density of states of the Landau levels in the space of energy and magnetic flux, which can be directly compared to the dI/dV measurement from the scanning tunneling microscopy (STM) and quantum oscillations from magneto-transport measurement. Please find the new figure 5 and the last paragraph in the discussion section in the revised manuscript.

Some additional comments:

a) Eq 5 is the main result used in this work, and the authors should provide a full derivation of it, whether that can be found elsewhere or not. It is also interesting to see why it does not have any dependence on ε_n , even though the mathematically equivalent expression (Eq 3) depends explicitly on ε_n . In other words, how can $E_{n,B}(k)$ "know" about ε_n , which can be shifted up or down at will, independent of the other bands?

Author's Response: We thank the reviewer for reading our manuscript carefully. In fact, $\varepsilon_m(\mathbf{k})$ in $E_{n,B}(\mathbf{k}) = -2\pi \frac{\phi}{\phi_0} \frac{1}{A_0} \text{Im} \sum_{m \neq n} \varepsilon_m(\mathbf{k}) \chi_{xy}^{nm}(\mathbf{k})$ should be interpreted as the energy of the m -th band with respect to the flat band energy because we assume that the flat band energy is zero. To clarify this point, we explicitly mentioned in the main text and Supplementary Information of the revised manuscript as "Since we assume that the flat band's energy is zero, one should interpret $\varepsilon_m(\mathbf{k})$ as the energy of the m -th band with respect to the flat band energy."

As for the question about the derivation of Eq.(5), please find our response to the reviewer’s question 2 above.

b) In Fig 1 and the discussion, it is clear that the modified band structure $E_{n,B}(k)$ does not give the actual energy levels of the actual Landau levels, even though it determines the energy range of the Landau levels. Is this interpretation correct? If yes, the authors should make that clear. Also, what determines the spacing between the Landau levels within the energy spread ?

Author’s Response: We thank the reviewer for this important comment. The actual Landau levels of the flat band can be obtained in principle by analyzing the modified band structure $E_{n,B}(k)$ using the Onsager’s semiclassical scheme. While the original Onsager’s scheme is described by

$$S_0(\epsilon) = \frac{2\pi eB}{\hbar} \left(n + \frac{1}{2} - \frac{\gamma_{\epsilon,B}}{2\pi} \right), \quad (11)$$

where $S_0(\epsilon)$ is the area of the semiclassical orbit on the original band structure $\epsilon_n(\mathbf{k})$ without the magnetic field, it was later shown that one can obtain more accurate results by using a modified band structure $E_{n,B}(k)$ instead of the original band structure. Here, the modified one is obtained by adding a correction term proportional to the magnetic field to the original band structure as shown below.

$$E_{n,B}(\mathbf{k}) = \epsilon_n(\mathbf{k}) + \mu_n(\mathbf{k})B. \quad (12)$$

In the case of the flat band, this B-linear correction becomes dominant and takes a crucial role in the semiclassical analysis as we found in our manuscript. While one cannot apply the conventional Onsager’s scheme to the flat band directly because the semiclassical orbits are not well-defined in such a dispersionless band, one can apply this semiclassical scheme to the modified band dispersion $E_{n,B}(k)$ because it is dispersive in general. Especially, around the band edges of $E_{n,B}(k)$, the modified dispersion can be approximated to a parabolic dispersion, where one can define the effective mass m^* . Then, the Onsager’s scheme predicts equally spaced Landau levels with the spacing $\hbar eB/m^*$. One interesting point to note is that m^* is inversely proportional to B because $\mu_n(\mathbf{k})B$ is proportional to B . As a result, the Landau level spacing from the modified band structure is proportional to B^2 .

To clarify this point, we added some sentences in the second paragraph of the section ‘Modified band dispersion and the Landau level spreading’ in the ‘Results’ part as follows. “As a result, one can obtain the Landau levels of the IFB in the adjacent gapped regions by applying the semiclassical quantization rule to $E_{n,B}(\mathbf{k})$, which naturally explains the LLS of the IFB. Especially, around the band edges of $E_{n,B}(\mathbf{k})$, one can define the effective mass m^* , which is inversely proportional to B , from which the Onsager’s scheme predicts Landau levels with a spacing $\hbar eB/m^* \propto B^2$. The resulting Landau spectrum is bounded by the upper and lower band edges of $E_{n,B}(\mathbf{k})$.”

REVIEWER COMMENTS

Reviewer #3 (Remarks to the Author):

In the revised manuscript, the authors have addressed most of my concerns, as well as the other referees'. Hence, the manuscript as it stands is much stronger than before.

My view is that the theoretical results are solid, but significance still has room to be better communicated:

1) In Supp Note 1, the authors have now derived Eq 5 from the definition of μ . But to broad audience, it is still unclear why Eq 3 should even hold - that is a known result, but since it is so central to the conclusions, its derivation should be explained starting from first principles too.

2) The connection to experiments can still be better explained. The authors had made efforts in discussing how the LL spread can look like in more realistic scenarios. But it will be good to further explain (i) what will the experimentalist actually see, and what should be concretely measured; (ii) are there existing experiments that point towards the proposed phenomena, (iii) if not, is it actually possible to observe this spreading in synthetic systems like photonics, electrical circuits, etc? The answer may be found from a first principle derivation as suggested above.

If yes, that will be a statement that greatly increases the impact of this work.

Reply to the third reviewer

In the revised manuscript, the authors have addressed most of my concerns, as well as the other referees'. Hence, the manuscript as it stands is much stronger than before. My view is that the theoretical results are solid, but significance still has room to be better communicated:

Author's Response: We indeed thank the reviewer for reading our manuscript carefully again and giving us valuable comments that were very helpful to improve our manuscript. We are also pleased to hear that our previous response resolved most of the reviewer's concerns. Below, we have addressed all of the reviewer's additional comments carefully. We hope the reviewer can find that our revised manuscript is suitable for publication in Nature Communications.

1) In Supp Note 1, the authors have now derived Eq. (5) from the definition of μ . But to broad audience, it is still unclear why Eq. (3) should even hold - that is a known result, but since it is so central to the conclusions, its derivation should be explained starting from first principles too.

Author's Response: We thank the reviewer for this suggestion. Following the reviewer's suggestion, we have added the detailed explanation and derivation of Eq. (3), starting from the first principles, in Supplementary Note 1. In the derivation of Eq. (3), the Peierls substitution and the semiclassical wave packet dynamics play the role of the first principles.

Here let us briefly explain how the modified band structure $E_{n,B}(\mathbf{k})$ in Eq. (3) is derived, rather than presenting the full details of the derivation, which can be found in Supplementary Note 1. The derivation starts from considering a wave packet that is localized well in both real and momentum spaces. The wave packet is made from the Bloch wave function of the n th band. When a magnetic field is turned on, the Hamiltonian changes according to the Peierls substitution, namely the substitution of the momentum operator \mathbf{p} for $\mathbf{p} - e\mathbf{A}(\mathbf{r})$ where $\mathbf{A}(\mathbf{r})$ is the vector potential that satisfies $\nabla_{\mathbf{r}} \times \mathbf{A}(\mathbf{r}) = \mathbf{B}$. Accordingly, the vector potential generates spatial variation of the Hamiltonian, and the wave packet experiences this spatial variation as the center of the wave packet moves. Such a spatial variation of the Hamiltonian can be treated as a perturbation near the wave packet's center, which has the form of the orbital magnetic moment. In this way, Eq. (3) and Eq. (4) can be obtained. More details are presented in Supplementary Note 1.

2) The connection to experiments can still be better explained. The authors had made efforts in discussing how the LL spread can look like in more realistic scenarios. But it will be good to further explain (i) what will the experimentalist actually see, and what should be concretely measured; (ii) are there existing experiments that point towards the proposed phenomena, (iii) if not, is it actually possible to observe this spreading in synthetic systems like photonics, electrical circuits, etc? The answer may be found from a first principle derivation as suggested above. If yes, that will be a statement that greatly increases the impact of this work.

Author's Response: We thank the reviewer for this constructive comment regarding the connection to experiments. We now answer the comments (i), (ii), (iii) in order.

(i) In our previous response, we computed the Landau level fan diagram of the disordered flat band model, which is a plot of the density of states (DOS) peaks as a function of the magnetic field. The

Landau fan diagram can be accessed experimentally by measuring several concrete quantities such as the longitudinal resistance from magnetotransport measurement or dI/dV values from scanning tunneling microscopy (STM) as explained in detail below.

First, in the magnetotransport measurement, the longitudinal resistance R_{xx} and transverse (Hall) resistance R_{xy} are measured in various ranges of the magnetic field B and carrier density n . The resistances are represented by a color map where the horizontal and vertical axes represent n and B , respectively. Then, the Landau fan diagram is obtained by identifying the peaks in R_{xx} . Conventionally, the gaps between the Landau levels are identified from the minima in R_{xx} . One typical example of the Landau fan diagram obtained from the measured data is shown in Fig. 1(a). In [X. Lu et al., PNAS **118** (30) e210006118 (2021)], the authors measured R_{xx} of twisted-bilayer graphene near the second magic angle and obtained the Landau fan diagram. The computed Landau fan diagram can be compared with the one obtained from experiments.

Figure 1: Experimental techniques for measuring the Landau levels. (a) Typical example of the Landau fan diagram, adopted from [X. Lu et al., PNAS **118** (30) e210006118 (2021)]. (b) Typical example of the dI/dV spectrum from the STM measurement, adopted from [H. Du et al., Nat. Commun. **7**, 10814 (2016)].

Second, the STM experiment is an experimental technique that can observe the Landau levels more directly. A bias voltage (V) is applied between the STM tip and the sample. When electrons tunnel between the tip and sample, the tunneling current (I) is generated. In low temperatures, the tunneling current I is proportional to the density of states (DOS) integrated up to V . For this reason, the dI/dV spectrum yields the local DOS. When the magnetic field is applied, the Landau levels are formed and can be observed as peaks in the dI/dV spectrum. A typical example of the dI/dV spectrum obtained by the STM measurement is shown in Fig. 1(b).

We expect the lowest/highest Landau level of an isolated flat band, which defines the magnitude of Landau level spreading, can be experimentally detected through these measurements. Even in disordered systems, the lowest and highest Landau levels can be successfully identified as long as the Landau level spacing becomes larger than the Landau level broadening induced by disorders. The Landau level spreading induces the shift of the DOS peak positions, which becomes more prominent when the Landau level spreading appears asymmetrically in the energy direction, as shown in the spin-orbit coupled Lieb lattice

model.

To stress this point, we have added some sentences in the last paragraph of the ‘Discussion’ part as follows. *“We expect the DOS peak corresponding to the LLL to be detected by the resistance measurement from magnetotransport experiments or the dI/dV measurement from the scanning tunneling spectroscopy if the magnetic field is strong enough or the system is sufficiently clean so that the Landau level spacing becomes larger than the Landau level broadening. Especially, when the LLS develops asymmetrically, like in Fig. 5(b), an overall energy shift of the DOS from the flat band’s energy appears more prominently, which provides a direct experimental signature of the LLS even in disordered systems.”*

(ii) Up to date, twisted-bilayer graphene and kagome materials such as FeSn and CoSn are the only synthesized materials exhibiting flat bands. There are several experimental studies on the Landau levels in twisted-bilayer graphene such as [X. Lu et al., Nature **574**, 653 (2019); Y. Saito et al., Nature Physics **17**, 478 (2021); X. Lu et al., PNAS **118** (30) e2100006118 (2021)]. Since twisted-bilayer graphene has a large unit cell, the high magnetic field required to obtain the detailed Landau fan diagram can be more easily reached than other materials. However, twisted-bilayer graphene exhibits degenerate flat bands with nontrivial band topology, which is not directly related to our work focusing on nondegenerate flat bands with trivial band topology.

As far as we can tell, other than twisted-bilayer graphene, there is no report of the experimental study on kagome materials related to the Landau levels of flat band yet.

(iii) The influence of the magnetic field can be implemented in a tight-binding Hamiltonian by properly attaching Peierls phases to the hopping amplitudes. This, at the same time, means that even an artificial magnetic field can be applied to lattice systems by suitably engineering the complex hopping amplitudes. In fact, there are various methods of implementing such artificial magnetic fields proposed up to now. For example, in [M. Aidelsburger et al., Phys. Rev. Lett. **107**, 255301 (2011)], it is shown that Raman-assisted tunneling generates artificial magnetic field in an optical lattice with cold atoms. Also, an array of resonators can realize the artificial magnetic field in photonic systems [K. Fang et al., Nat. Photonics **6**, 782 (2012); M. Hafezi et al., Nat. Photonics **7**, 1001 (2013)]. Moreover, even synthetic dimensions can be used to implement artificial magnetic field as shown in [M. Mancini et al., Science **349**, 1510 (2015); B.K. Stuhl et al., Science **349**, 1514 (2015)] where the hyperfine states of atoms serve the synthetic dimension. Various experimental studies on artificial magnetic fields are discussed in a recent review article [T. Ozawa et al., Rev. Mod. Phys. **91**, 015006 (2019)]. Considering that the realization of artificial magnetic fields has become a very active field, we believe that the synthetic systems discussed above provide promising platforms to realize the isolated flat band models proposed in our work.

Regarding this point, we have added some sentences in the last paragraph of the ‘Discussion’ part as follows. *“Up to now, our discussion has been focused on conventional materials to realize flat bands. However, it is worth noting that there are various artificial systems such as photonic systems, optical lattices, and systems with synthetic dimensions, which could offer better opportunities to test our theoretical prediction. In these systems, the band engineering is relatively easier, and controlled experiments with artificial magnetic fields can also be performed. Designing realistic experimental setups for observing LLS of flat bands in such artificial systems would be one important problem for future study.”*

Finally, we note that strain can also generate (pseudo-) magnetic field. For example, it was shown in [M.I. Katsnelson, K.S. Novoselov, Solid State Commun. **143** 3 (2007)] that strain on the graphene induces

Figure 2: Flat band model in the honeycomb lattice with strain-induced artificial magnetic field. (a) The lattice model with a finite size along the vertical direction. The hopping parameter $t_1(R_y)$ varies spatially because of the strain: $t_1(R_y) = t_1^{(0)}[1 + \lambda(R_y - N_y/2)]$. (b) Band structure without the strain. The K and K' points in 2D Brillouin zone (BZ) are projected to $k_x = -2\pi/3$ and $2\pi/3$ in the surface BZ. Surface states are denoted by red lines. (c) Band structure with the strain ($\lambda = 0.01$). (d) Zoom-in view of the flat band in (c) where the LLS can be observed. In (a)-(c), $N_x = 21$, $e_1 = 1$, $e_2 = 3$, and $t_1^{(0)} = t' = 1$ are used in the calculations.

a change of hopping parameters, and modifies the Dirac Hamiltonian describing the band structure of graphene near $K = (4\pi/3, 0)$ and $K' = (-4\pi/3, 0)$ in a way that $\mathbf{k} \cdot \boldsymbol{\sigma}$ is replaced by $(\mathbf{k} - \frac{e}{\hbar}\mathbf{A}) \cdot \boldsymbol{\sigma}$. To address the reviewer's comment more thoroughly, we made an effort to implement this idea into a flat band system as shown below. Our method closely follows [G. Salerno et al., 2D Mater., **2**, 034015 (2015); O. Jamadi et al., Light: Science & Applications **9**, 144 (2020)], where a spatially varying hopping is shown to create strain-induced Landau levels in the honeycomb lattice.

In the honeycomb lattice with the nearest-neighbor hopping $t_1^{(0)} = 1$, the band structure exhibits two Dirac cones at $K = (4\pi/3, 0)$ and $K' = (-4\pi/3, 0)$. When the second nearest-neighbor hoppings $t' = 1$ between B sublattices and on-site potential $e_1 = 1$ for A sublattice and $e_2 = 3$ for B sublattice are considered additionally, an isolated flat band can be obtained [Fig. 2(b)]. To apply strain, along with open boundary condition in y -direction, we consider spatially varying hopping parameter, $t_1(R_y) = t_1^{(0)}[1 + \lambda(R_y - N_y/2)]$, as shown in Fig. 2(a). Note that the continuum Hamiltonian near K and K' points are given by

$$H_K(\mathbf{k}) = -\frac{\sqrt{3}}{2}t(k_x - eA_{x,K})\sigma_x + \frac{\sqrt{3}}{2}tk_y\sigma_y + \text{Diag}(t^2, 0), \quad (1)$$

$$H_{K'}(\mathbf{k}) = \frac{\sqrt{3}}{2}t(k_x - eA_{x,K'})\sigma_x + \frac{\sqrt{3}}{2}tk_y\sigma_y + \text{Diag}(t^2, 0). \quad (2)$$

where $A_{x,K} = A_{x,K'} = \frac{2}{\sqrt{3}}\frac{\hbar}{e}\lambda y$. Hence, (pseudo-) magnetic flux per unit cell is $\phi/\phi_0 = \pm\lambda/(2\pi)$. Note that signs of magnetic flux near K and K' are opposite. Comparing the band structures without strain (Fig. 2(b)) and with strain (Fig. 2(c)), one can clearly observe that the flat band acquires finite bandwidth under strain.

However, there is a critical limitation of this approach. That is, the overall bandwidth of the flat band is an order of $\lambda N_y t_1^{(0)}/2$, while the predicted LLS due to the pseudo-magnetic field is an order of $\lambda t_1^{(0)}$. Such a large bandwidth appears because a flat band generally develops finite band dispersion when hopping amplitudes are modified, which breaks the flat-band condition at the same time. It is worth noting that a flat band appears when the hopping amplitudes are suitably fine-tuned in a given lattice. Once such a fine-tuning condition is violated, the flat band quickly develops a dispersion. This type of the problem always occurs when pseudo-magnetic field is implemented in flat band systems in synthetic systems. Therefore

careful designing of the modulated hopping amplitudes, which implements the pseudomagnetic field while preserving the flat band condition, is required. We believe that designing a realistic setup to realize the LLS of flat bands in synthetic systems is definitely an important problem but goes beyond the scope of the present work. Thus we leave this problem for future study.